# Activation of Anti-SARS-CoV-2 Human CTLs by Extracellular Vesicles Engineered with the N Viral Protein

**DOI:** 10.3390/vaccines10071060

**Published:** 2022-06-30

**Authors:** Francesco Manfredi, Chiara Chiozzini, Flavia Ferrantelli, Patrizia Leone, Andrea Giovannelli, Massimo Sanchez, Maurizio Federico

**Affiliations:** 1National Center for Global Health, Istituto Superiore di Sanità, Viale Regina Elena 299, 00161 Rome, Italy; francesco.manfredi@iss.it (F.M.); chiara.chiozzini@iss.it (C.C.); flavia.ferrantelli@iss.it (F.F.); patrizia.leone@iss.it (P.L.); 2National Center for Animal Experimentation and Welfare, Istituto Superiore di Sanità, Viale Regina Elena 299, 00161 Rome, Italy; andrea.giovannelli@iss.it; 3Core Facilities, Istituto Superiore di Sanità, Viale Regina Elena 299, 00161 Rome, Italy; massimo.sanchez@iss.it

**Keywords:** extracellular vesicles, SARS-CoV-2, CTL immunity, HIV-1 Nef, CD107a

## Abstract

We propose an innovative anti-SARS-CoV-2 immune strategy based on extracellular vesicles (EVs) inducing an anti-SARS-CoV-2 N CD8^+^ T cytotoxic lymphocyte (CTL) immune response. We previously reported that the SARS-CoV-2 N protein can be uploaded at high levels in EVs upon fusion with Nef^mut^, i.e., a biologically inactive HIV-1 Nef mutant incorporating into EVs at quite high levels. Here, we analyze the immunogenic properties in human cells of EVs engineered with SARS-CoV-2 N fused at the C-terminus of either Nef^mut^ or a deletion mutant of Nef^mut^ referred to as Nef^mut^PL. The analysis of in vitro-produced EVs has supported the uploading of N protein when fused with truncated Nef^mut^. Mice injected with DNA vectors expressed each fusion protein developed robust SARS-CoV-2 N-specific CD8^+^ T cell immune responses. When ex vivo human dendritic cells were challenged with EVs engineered with either fusion products, the induction of a robust N-specific CTL activity, as evaluated by both CD107a and trogocytosis assays, was observed. Through these data we achieved the proof-of-principle that engineered EVs can be instrumental to elicit anti-SARS-CoV-2 CTL immune response in human cells. This achievement represents a mandatory step towards the upcoming experimentations in pre-clinical models focused on intranasal administration of N-engineered EVs.

## 1. Introduction

The systemic administration of current vaccines against respiratory viruses can result in a temporary immunity, mostly based on the production of specific antibodies [1,2]. Besides antibody waning, their major limitation consists of a fleeting to undetectable immune memory at the viral port of entry, i.e., the mucosa of high- and low-respiratory tracts, where B-memory cells were found rarely despite their presence in the blood of vaccinees [3,4,5,6]. This defect can be hardly overcome since lung cell immunity is largely not influenced by that developed in peripheral circulation and lymphoid organs. Lymphocytes in lungs are maintained independently of the pool of circulating lymphocytes, and their continuous loss through intraepithelial migration towards airways is replenished by homeostatic proliferation. This applies to CD8^+^ T lymphocytes [7,8] and, as demonstrated by recent findings, B lymphocytes also. On this subject, CXCR3^+^ CD62L^−^ lung-resident memory B cells (BRMCs) were described as the major memory effectors of humoral antiviral immunity [9]. Based on these findings, blocking virus replication at the mucosa of upper respiratory tract represents a key step in the fight against respiratory viruses [10].

Although a robust CD8^+^ T-cell response is strongly beneficial in the fight against SARS-CoV-2 infection [11], no vaccine technology specifically devoted to the induction of this branch of adaptive immunity is currently available for humans. Differently to what is observed with the humoral immunity, SARS-CoV-2-specific CD8^+^ T cell immunity is expected to maintain intact its efficacy in the presence of the amino acid substitutions occurring in emerging viral variants [12,13]. In addition, based on data obtained with patients recovered from SARS-CoV infection, CD8^+^ T cell antiviral immunity is expected to last more than a decade [14].

We developed an original strategy to induce CTL immunity based on the in vivo engineering of extracellular vesicles (EVs). All cell types constitutively release EVs nanovesicles, which are key players of intercellular communication [15]. When produced by healthy cells, EVs comprise exosomes and microvesicles. Exosomes are lipid bilayer vesicles of 50–200 nm in diameter and form intracellularly upon inward invagination of endosome membranes. Intraluminal vesicles (ILVs) produced in this way form multivesicular bodies (MVBs) that can be trafficked to plasma membrane and released. Microvesicles are up to 1000 nanometers in diameter and show both physical and biochemical features similar to those of exosomes, albeit being generated through direct extrusion of plasma membrane.

We previously identified a biologically inactive Human Immunodeficiency Virus (HIV)-Type 1 Nef protein (Nef^mut^) having an unusually high efficiency of incorporation into EVs even when foreign polypeptides are fused to its C-terminus [16,17]. After the intramuscular (i.m) injection of DNA vectors coding for either viral or tumor antigens fused with Nef^mut^, high amounts of the fusion protein are uploaded in muscle cells-derived EVs and are supposed to be internalized by tissue-resident antigen-presenting cells (APCs). These cells cross-present EV contents to activate antigen-specific CD8^+^ T cells. In addition, these “in vivo” engineered EVs are expected to freely circulate into the body, thus having the potential to reach distal tissues.

We recently demonstrated both immunogenicity and efficacy in mouse DNA vectors expressing diverse SARS-CoV-2 structural antigens fused with Nef^mut^ and injected intramuscularly in mice [18,19]. To validate a novel strategy of immunization at the viral port of entry based on engineered EVs, we here evaluated the CTL immunogenicity of purified EVs engineered with SARS-CoV-2 N fused with either full-length Nef^mut^ or a C-terminal truncated form of it (referred to as Nef^mut^PL) [20], whose characteristics could represent an improvement in terms of the safety profile of the immunogen.

## 2. Materials and Methods

### 2.1. DNA Vector Synthesis

Nef^mut^, Nef^mut^/N and Nef^mut^PL open reading frames (ORFs) were cloned into pVAX1 plasmid (Thermo Fisher, Waltham, MA, USA), i.e., a vector approved by FDA for use in humans, as previously described [14,15,16]. ORFs coding for Nef^mut^PL fused with SARS-CoV-2 N protein was cloned into pVAX1 plasmid as well. To this aim, an intermediate vector referred to as pVAX1/Nef^mut^fusion was used [18]. Here, the Nef^mut^PL ORF deprived of its stop codon was followed by a sequence coding a GPGP linker, including a unique Apa I restriction site. In this way, sequences comprising the Apa I site at their 5′ end, and the Pme I one at their 3′ end were fused in frame with Nef^mut^PL ORF. Stop codons was preceded by sequences coding for a DYKDDDK epitope tag (flag-tag). The vector was synthesized by Explora Biotech.

### 2.2. Cell Cultures and Transfection

Both human embryonic kidney (HEK) 293T cells (ATCC, CRL-11268) and human HLA-A02 MCF-7 cells (ATCC, HTB-22) were grown in DMEM (Gibco, Thermo Fisher, Waltham, MA, USA) plus 10% heat-inactivated fetal calf serum (FCS, Gibco, Thermo Fisher, Waltham, MA, USA). Transfection assays were performed using Lipofectamine 2000 (Invitrogen, Thermo Fisher Scientific, Waltham, MA, USA)-based method. Monocytes were isolated from peripheral blood mononuclear cells (PBMCs) of HLA-A.02 healthy donors by immune magnetic protocol. Human immature dendritic cells (iDCs) were obtained after 5 to 7 days of the culture of monocytes in the presence of both IL-4 and GM-CSF as described [21]. The iDCs were routinely characterized by fluorescence-activated cell sorting (FACS) analysis for the expression of CD1a, CD14, CD80, CD83, and CD86 markers. All required antibodies were from eBio, Thermo Fisher Scientific. DC maturation was obtained through overnight treatment with 10 ng/mL of lipopolysaccharide (LPS) and verified by FACS analysis.

### 2.3. EV Isolation

HEK293T cells transfected with vectors expressing the Nef^mut^-based fusion proteins were washed 24 h later and reseeded in medium supplemented with EV-deprived FCS. Supernatants were harvested from 48 to 72 h after transfection. EVs were recovered through differential centrifugations [22] by centrifuging supernatants at 500× *g* for 10 min, and then at 10,000× *g* for 30 min. Supernatants were harvested, filtered with 0.22 μm pore size filters, and ultracentrifuged at 70,000× *g* for l h. Pelleted vesicles were resuspended in 1×PBS, and ultracentrifuged again at 70,000× *g* for 1 h. Afterwards, pellets containing EVs were resuspended in 1:100 of the initial volume.

### 2.4. Western Blot Analysis

Western blot analyses of both cell lysates and EVs were carried out after resolving samples in 10% sodium dodecyl sulfate-polyacrylamide gel electrophoresis (SDS-PAGE). The analysis on cell lysates was carried out by washing cells twice with 1×PBS (pH 7.4) and lysing them with 1× SDS-PAGE sample buffer. Samples were resolved by SDS-PAGE and transferred by electroblotting on a 0.45 μm pore size nitrocellulose membrane (Amersham) overnight using a Bio-Rad Trans-Blot. EVs were lysed and analyzed as described for cell lysates. For immunoassays, membranes were blocked with 5% non-fat dry milk in PBS containing 0.1% Triton X-100 for 1 h at room temperature, then incubated ON at 4 °C with specific antibodies diluted in PBS containing 0.1% Triton X-100. Filters were revealed using 1:1000-diluted sheep anti-Nef antiserum ARP 444 (a generous gift of M. Harris, University of Leeds, UK), 1:500-diluted anti-β-actin AC-74 mAb from Sigma (St. Louis, MO, USA), 1:500 diluted anti-CD81 monoclonal Ab cat. N.10616D from Thermo Fisher Scientific were analyzed by a Chemi-Doc apparatus, Bio-Rad, and relevant signals were quantified by Image Lab software version 6.1.

### 2.5. Mice Immunization

Six-weeks old C57 Bl/6 were obtained from Charles River. They were hosted at the Central Animal Facility of the Istituto Superiore di Sanità (ISS), as approved by the Italian Ministry of Health, authorization n. 565/2020 released on 3 June 2020. The day before the first inoculation, microchips from DATAMARS were inserted sub cute at the back of neck between the shoulder blades on the dorsal midline. Preparations of 10 μg for each DNA vector were diluted in 30 μL of sterile 0.9% saline solution. Both quality and quantity of the DNA preparations were checked by 260/280 nm absorbance and electrophoresis assays. Mice were anesthetized with isoflurane as prescribed in the Ministry authorization. Each inoculum volume was, therefore, measured by micropipette, loaded singly into a 1 mL syringe without dead volume, and injected into mouse quadriceps. Immediately after inoculation, mice underwent electroporation at the site of injection through the Agilpulse BTX device using a 4-needle array 4 mm gap, 5 mm needle length, with the following parameters: 1 pulse of 450 V for 50 µs; 0.2 ms interval; 1 pulse of 450 V for 50 µs; 50 ms interval; 8 pulses of 110 V for 10 ms with 20 ms intervals. The same procedure was repeated for both quadriceps of each mouse. Immunizations were repeated after 14 days. Fourteen days after the second immunization, mice were sacrificed by either cervical dislocation or CO_2_ inhalation, following the recommendations included in the Ministry authorization protocol.

### 2.6. Cell Isolation from Immunized Mice

Spleens were explanted by qualified personnel of ISS Central Animal Facility and transferred into a 60 mm Petri dish containing 2 mL of RPMI 1640 (Gibco, Thermo Fisher, Waltham, MA, USA), 50 µM 2-mercaptoethanol (Sigma, St. Louis, MO, USA). Splenocytes were extracted by incising the spleen with sterile scissors and pressing the cells out of the spleen sac with the plunger seal of a 1 mL syringe. After addition of 2 mL of RPMI medium, cells were transferred into a 15 mL conical tube, and the Petri plate was washed with 4 mL of medium to collect remaining cells. After a three-minute sedimentation, splenocytes were transferred to a new sterile tube to remove cell/tissue debris. Counts of live cells were carried out by the trypan blue exclusion method. Fresh splenocytes were resuspended in RPMI complete medium, containing 50 µM 2-mercaptoethanol and 10% FBS, and tested by IFN-γ EliSpot assay.

### 2.7. IFN-γ EliSpot Analysis

A total of 2.5 × 10^5^ live cells were seeded in each IFN-γ EliSpot microwell. Cultures were run in triplicate in EliSpot multiwell plates (Millipore, Burlington, MA, USA, cat n. MSPS4510) pre-coated with the AN18 mAb against mouse IFN-γ (Mabtech, Nacka Strand, Sweden) in RPMI 1640 (Gibco, Thermo Fisher, Waltham, MA, USA), 10% FCS, 50 µM 2-mercaptoethanol (Sigma, St. Louis, MO, USA) for 16 h in the presence of 5 µg/mL of the CD8-specific SARS-CoV-2 N (H2-K^b^) 219–228 ALALLLLDRL [23] or HIV-1 Nef (H2-K^b^) 48–56 TAATNADCA peptides [24]. As negative control, 5 µg/mL of a H2-K^b^-binding peptide were used. For cell activation control, cultures were treated with 10 ng/mL phorbol 12-myristate 13-acetate (PMA, Sigma) plus 500 ng/mL of ionomycin (Sigma). After 16 h, cultures were removed, and wells incubated with 100 µL of 1 µg/mL of the R4-6A2 biotinylated anti-IFN-γ (Mabtech) for 2 h at r.t. Wells were then washed and treated for 1 h at r.t. with 1:1000 diluted streptavidine-ALP preparations from Mabtech. After washing, spots were developed by adding 100 µL/well of SigmaFast BCIP/NBT, Cat. N. B5655. Spot-forming cells were finally analyzed and counted using an AELVIS EliSpot reader.

### 2.8. NTA Analysis of Nanovesicles

For nanoparticle tracking analysis (NTA), purified EVs were diluted in PBS and assayed with a Nanosight NS300 with the NTA software (Malvern Panalytical Ltd., Malvern, UK) through a 488 nm laser.

### 2.9. Cross-Priming Assay

Immature (i)DCs were challenged by engineered EVs uploading either Nef^mut^, Nef^mut^/N, or Nef^mut^PL/N isolated from supernatants of HEK293T transfected cells. After overnight incubation, iDCs were matured by LPS treatment for 24 h. Thereafter, DCs were washed, and co-cultured with autologous peripheral blood lymphocytes (PBLs) in a 1:10 cell ratio. A week later, the stimulation procedure was repeated, and, after an additional week, CD8^+^ T cells were recovered for downstream assays.

### 2.10. Activation-Induced Degranulation and Trogocytosis Assays

Activation-induced degranulation was measured by evaluating CD107a surface expression as described [25]. Briefly, 2 × 10^5^ PBLs recovered from cross-primed cultures were co-cultivated for 5 h with equal number of HLA-matched target cells, i.e., MCF-7 previously treated with 1 µg/mL of either SARS-CoV-2 N-specific or unrelated HLA-A.02 binding peptides, in the presence of both PE-conjugated anti-CD107a mAb H4A3 (BD-Pharmingen) and 0.7 µg/mL monensin (GolgiStop, BD, Biosciences, Franklin Lakes, NJ, USA). The gated CD8^+^ T cell populations were then analyzed by FACS for the detection of CD107a-related fluorescence.

For trogocytosis assay, MCF-7 cells were labeled with the fluorescent lipophilic dye CM-Dil (Molecular Probes, Invitrogen Thermo Fisher, Waltham, MA, USA) according to the manufacturer’s instructions with minor modifications. In detail, 10^6^ target cells were resuspended in 1 mL of 1×PBS in the presence of 1 μM CM-Dil and incubated for 5 min at 37 °C followed by an additional incubation of 15 min at 4 °C. Cold RPMI medium containing 10% AB human serum was added to stop labeling (1:1 volume) for 10 min at 4 °C. Then, cells were washed three times with 1×PBS containing 1% AB human serum. Following resuspension in complete medium, labeled target cells were cultured for 4 h in the presence of 1 µg/mL of either SARS-CoV-2 N-specific or unrelated HLA-A.02 binding peptides, and finally co-cultured with primed PBLs (5 × 10^5^ per well in 200 μL of total volume) in U-shaped 96-well plates in a 1:5 ratio. After incubation of 5 h at 37 °C, cells were washed twice in 1×PBS containing 0.5 mM EDTA to ensure cell dissociation, resuspended in 1×PBS supplemented with 0.5% bovine serum albumin, and stained with both FITC-conjugated anti-human CD3 and APCCy7-conjugated anti-human CD8 Abs (both from BD-Pharmingen) for FACS analysis. Dead cells were stained by adding Aqua LIVE/DEAD dye.

For both assays, HLA-A.02-binding SARS-CoV-2 N-specific peptides were: 159–167 LQLPQGTTL, 219–227 ALLLLDRL, 316–324 GMSRIGMEV, and 351–359 ILLNKHIDA [26].

### 2.11. Statistical Analysis

When appropriate, data are presented as mean + standard deviation (SD). In some instances, the Mann–Whitney U test was used. *p* < 0.05 was considered significant.

## 3. Results

### 3.1. EV Uploading of SARS-CoV-2 N upon Fusion with Nef^mut^/PL

We previously demonstrated that cells expressing SARS-CoV-2 N protein fused with full-length Nef^mut^ release EVs uploaded with high amounts of the fusion product [18]. In the perspective of use in humans, reducing the size of the Nef^mut^ anchoring protein would be of safety relevance. The Nef^mu^ C-terminal truncation was already proven to not affect the EV uploading of HPV-E6,-E7 and SARS-CoV-2 S viral proteins [20]. To evaluate whether the C-terminal deletion of Nef^mut^ affects the N-uploading efficiency in EVs, both intracellular accumulation and association with EV of Nef^mut^PL/N were analyzed compared to that of Nef^mut^/N. To this aim, HEK293T cells were transiently transfected with the respective expression vectors and cell lysates analyzed 48 h thereafter. Meanwhile, EVs released on supernatants were collected 48–72 h after transfection, processed by differential centrifugations, and analyzed for the presence of Nef-related products by western blot (Figure 1 and Appendix A). Cell-associated steady-state levels of the Nef^mut^-derivatives appeared similar, and both fusion products are uploaded in EVs. As indicated by the densitometry analysis, a stronger efficiency of EV uploading was observed when SARS-CoV-2 N was fused with full-length Nef^mut^.

We concluded that SARS-CoV-2 N can be engineered in EVs upon fusion with either Nef^mut^ or Nef^mut^PL.

### 3.2. SARS-CoV-2 N-Specific Immunogenicity Induced in Mice Injected with DNA Vectors Expressing Either Nef^mut^/N or Nef^mut^PL/N

Next, we were interested in establishing whether the deletion of the EV anchoring protein affects the N-specific CD8^+^ T cell immunogenicity in mice. To this end, we compared the CD8^+^ T cell immune response induced in C57Bl/6 mice by injection of DNA vectors expressing either Nef^mut^/N or Nef^mut^PL/N. The read-out consisted of γ-IFN ELISpot analyses that were carried out using the H2-K^b^-restricted SARS-CoV-2 N- [23] or HIV-1 Nef-specific peptides [24]. The results indicated that both vectors were immunogenic. Consistently with the observed differences in the EV uploading efficiencies, a stronger CD8^+^ T cell immune response was observed in mice injected with the vector expressing full-length Nef^mut^ (Figure 2). On the other hand, the Nef-specific CD8^+^ T cell immune response appeared less potent throughout.

Hence, whatever the EV-anchoring protein considered, EVs engineered with SARS-CoV-2 N elicit an N-specific CD8^+^ T cell immune response.

### 3.3. SARS-CoV-2 N Specific CTLs Are Generated upon Challenge of Human DCs with Engineered EVs

Focus of the present study was obtaining the proof-of-principle on the ability of SARS-CoV-2 N engineered EVs to induce a specific CTL activity in human cells. To this aim, PBMCs from HLA-A02 healthy donors were tested in cross-priming assays carried out by co-cultivating human PBLs with autologous DCs previously challenged with EVs engineered with SARS-CoV-2 N.

In detail, human iDCs were challenged with similar amounts, as evaluated by the EV quantification carried out through NTA assay (Figure 3), of EVs engineered by either Nef^mut^, Nef^mut^/N or Nef^mut^PL/N.

After overnight culture in the presence of LPS, DCs (Appendix A) were put in co-culture with autologous PBLs for a week. Afterwards, the challenge was repeated and, after an additional week, the presence of SARS-CoV-2-specific CTLs was tested in two ways, i.e., through CD107a and trogocytosis assays.

The N-specific cytolytic activity of CD8^+^ T lymphocytes was first assessed through FACS analysis of CD107a/LAMP-1, i.e., a well characterized cell membrane marker of CTL degranulation [25]. To this end, lymphocytes recovered after cross-priming cultures were co-cultivated for five hours at both 1:1 and 5:1 cell ratios with syngeneic cells (i.e., MCF-7) previously treated with either unrelated or N-specific peptides. We noticed an increase in CD107a expression within the CD8^+^ T cells from PBLs co-cultivated with DCs challenged with N-engineered EVs compared to those co-cultivated with DCs incubated with control EVs (Figure 4). The increase appeared more sustained in the case of PBLs isolated from co-cultures with iDCs challenged with EVs engineered with Nef^mut^/N compared to those from DCs challenged with Nef^mut^PL/N EVs.

Antigen-specific CTLs can capture plasma membrane fragments from target cells while exerting the cytotoxic activity through a phenomenon referred to as trogocytosis [27]. The membrane capture is T-cell receptor dependent [28,29,30], epitope specific [30,31], and is exerted by lymphocyte clones endowed with the highest cytotoxic functions [32].

At the end of cross-priming cultures, PBLs were put in co-culture with peptide-treated MCF-7 cells previously labeled with CM-Dil, i.e., a red-fluorescent molecule specifically intercalating within cell membrane bilayers. After 5 h of incubation, the co-cultures were analyzed by FACS for the presence of CD8^+^/Dil^+^ T lymphocytes. As represented in Figure 5, a strong increase in red-fluorescent CD8^+^ T cells compared with control condition can be recognized in PBLs co-cultivated with DCs challenged with N-engineered EVs, indicating the actual presence of N-specific CTLs. Consistently with what was obtained with the CD107a assay, higher percentages of CD8^+^ T red-fluorescent cells were observed within PBLs co-cultivated with DCs challenged with Nef^mut^/N-engineered EVs compared to those co-cultivated with DCs treated with Nef^mut^PL/N EVs.

Together, these data offer the proof-of-principle that EVs engineered with SARS-CoV-2 N antigen have the potential to elicit an antigen-specific CTL immune response in humans. This finding is of obvious relevance in view of a possible translation into the clinic of a strategy of anti-SARS-CoV-2 immunization at the viral port of entry based on Nef^mut^-based EVs.

## 4. Discussion

The aim of present investigations was achieving the proof-of–concept on CTL immunogenicity in human cells of EVs engineered with the SARS-CoV-2 N protein. Demonstrating the effectiveness of such a tool would open the way towards innovative clinic applications devoted to the induction of immunity at the port of entry of SARS-CoV-2, as well as additional respiratory viruses.

Current anti-SARS-CoV-2 vaccines have limited efficacy in inducing virus-specific CD8^+^ T cell immunity. In addition, in view of the minimal passage of immune cells from peripheral circulation into respiratory tissues, the limited amounts of circulating virus-specific B-memory cells present in vaccinated subjects result in a virtual absence of a suitable antiviral B-cell memory at the respiratory mucosa.

For all these reasons, a strategy focused on the induction of immunity at the oral mucosa is anticipated to result in a more effective shield against respiratory viruses. We here show that the treatment of DCs with EVs engineered with SARS-CoV-2 N leads to the formation of antiviral CTLs in human cells. As already demonstrated for a plethora of both viral and tumor antigens [17], the whole SARS-CoV-2 N protein can be engineered in EVs upon fusion of both full length and, even with a somewhat reduced efficiency, C-truncated Nef^mut^. Notably, reduced EV incorporation in the presence of Nef^mut^/PL as anchoring protein was previously not observed with other viral proteins, including both S1 and S2 from SARS-CoV-2 [19]. It is possible that in the case of SARS-CoV-2 N protein, the already documented contribution to Nef uploading in nanovesicles of the Nef C-terminal domain [33,34] critically influences the EV association the fusion product. Notably, the lower EV uploading efficiency of N when fused with Nef^mut^PL paralleled with an apparent reduction in immunogenicity in mice, as well as in cross-priming assays carried out with human cells compared to that induced by N fused with full-length Nef^mut^. These pieces of experimental evidence support the idea that the efficiency of uploading in EVs as measurable “in vitro” through transfection experiments can predict the potency of the antigen-specific CD8^+^ T cell immune activation “in vivo”. A limitation of the present work is represented by the analysis of CTL immunogenicity that was limited to cells from HLA-A02 donors. Reproducing the investigations on cells with alternative HLAs would offer a more precise anticipation on the potential usefulness of engineered EVs as anti-SARS-CoV-2 immunogen.

Recent technologic advances allow large-scale production of EVs [35,36]. On the other hand, and as for most advanced experimental anti-Flu vaccines [37,38], the aerosol-mediated application of engineered EVs would ensure their encounter with mucosal dendritic cells (DCs) capable of sensing external stimuli and mounting appropriate local responses. These DCs can cross-present the contents of engineered EVs thereby initiating the CTL selection and activation in view of already described co-stimulatory signals provided by Nef^mut^ [39], which are conserved in Nef^mut^PL. Notably, the local application of engineered EVs would limit safety-based concerns in the use of EVs generated by cell lines. In sum, the potential advantages of this proposed vaccine strategy include recognition of all variants (considering the quite low mutation rate of the N gene), very few potential adverse events, immunity at the viral port of entry, and, on the basis of data published on SARS-CoV survivors [14], long lasting immunity.

Finally, the use of engineered EVs can be envisioned in combination with additional, nanoparticle-based immunogens able to induce antiviral humoral immunity and most desirably mucosal IgAs. In this way, through simple and safe aerosol administrations, a complete, as well as effective immunity at the viral port of entry could be attained. In the case of SARS-CoV-2, this strategy presents invaluable advantages compared to current vaccines.

## Figures and Tables

**Figure 1 vaccines-10-01060-f001:**
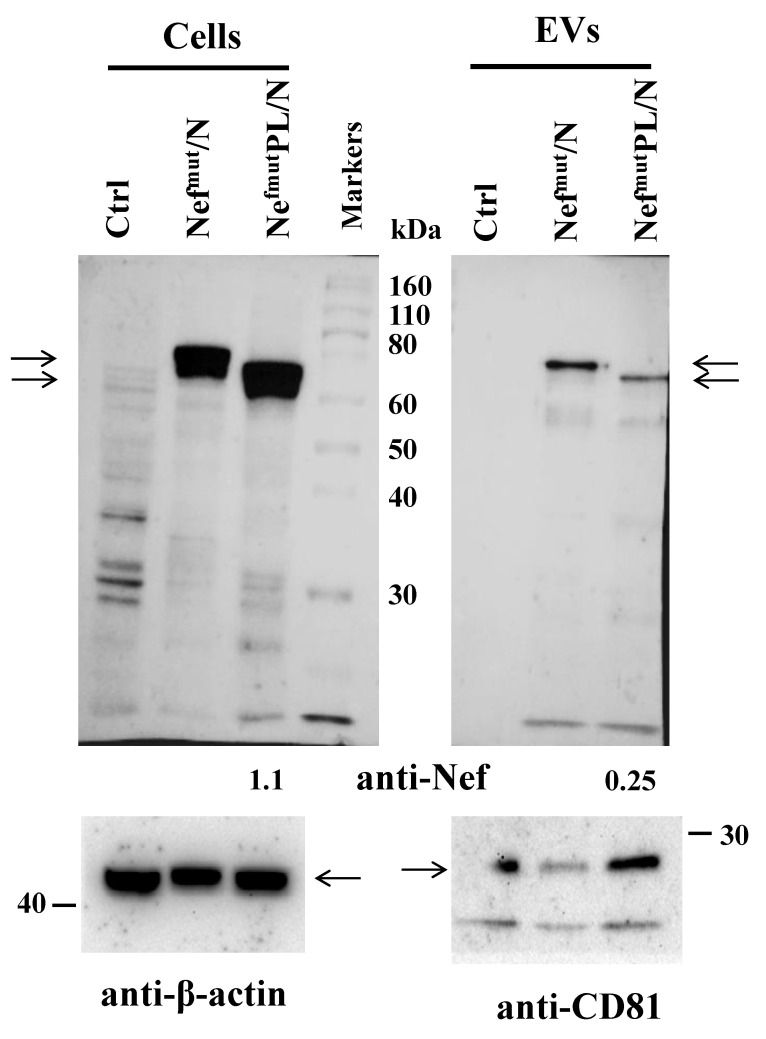
Detection of Nef^mut^-related products in cells and EVs after DNA transfection. Western blot analysis on 30 μg of lysates from HEK293T cells transfected with DNA vectors expressing either Nef^mut^/N, Nef^mut^PL/N, or, as control, void pVAX1 vector (Ctrl) (left panels). Equal volumes of buffer where purified EVs were resuspended after differential centrifugations of the respective cell supernatants were also analyzed (right panels). Polyclonal anti-Nef Abs served to detect Nef^mut^-based products, while β-actin and CD81 were revealed as markers for cell lysates and EVs, respectively. Molecular markers are given in kilodaltons (kDa). Arrows sign relevant products. Intensity ratios between Nef^mut^PL/N and Nef^mut^/N signals as revealed by densitometry analysis are also indicated. The results are representative of three independent experiments.

**Figure 2 vaccines-10-01060-f002:**
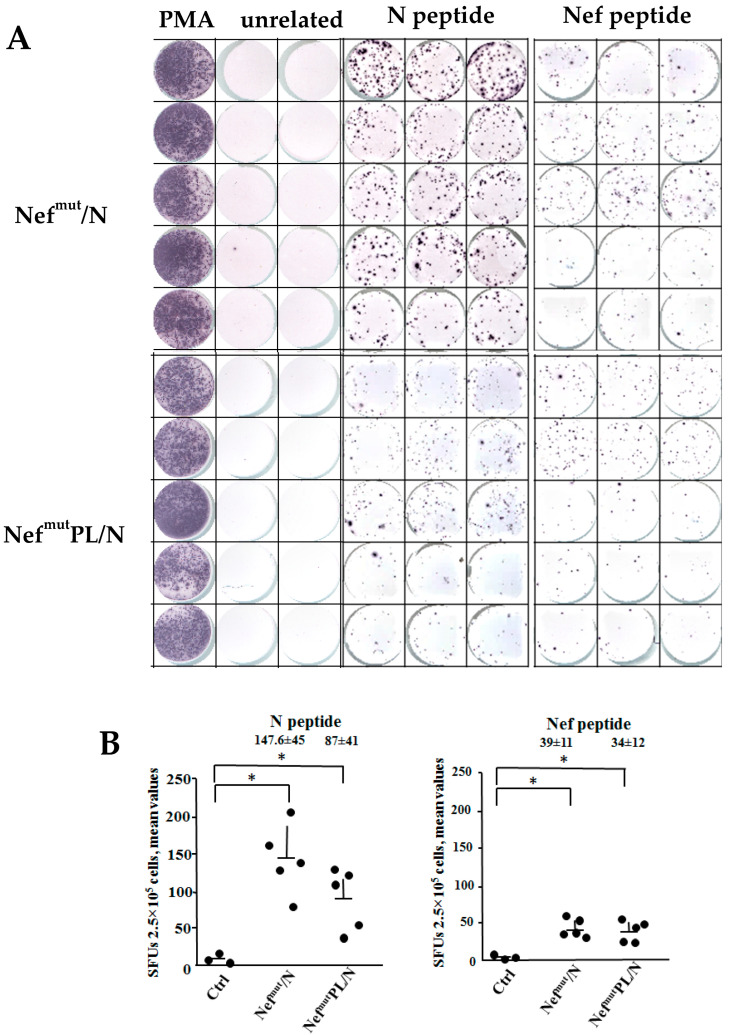
SARS-CoV-2-specific CD8^+^ T cell immunity induced in DNA injected mice. CD8^+^ T cell immune response in C57 Bl/6 mice inoculated i.m. with DNA vectors expressing either Nef^mut^/N, Nef^mut^PL/N or, as control, pVAX1. At the time of sacrifice, 2.5 × 10^5^ splenocytes were incubated ON with or without 5 µg/mL of either unrelated, SARS-CoV-2-N, or HIV-1 Nef-specific peptides in replicate IFN-γ EliSpot microwells. In panel (**A**), the raw data obtained with splenocytes from mice injected with either Nef^mut^/N or Nef^mut^PL/N are reported. PMA indicates cell cultures treated with PMA plus ionomycin. In panel (**B**), shown are the numbers of IFN- γ SFUs/well as mean values of triplicates after subtraction of mean values measured in wells seeded with splenocytes treated with unspecific peptides. Intragroup mean values + SD are also shown. The results are representative of two independent experiments. * *p* < 0.05.

**Figure 3 vaccines-10-01060-f003:**
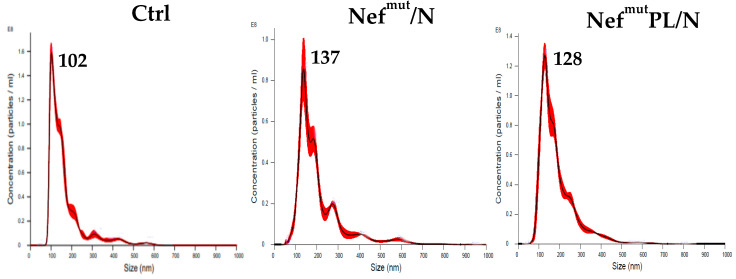
NTA analysis of EVs from transfected HEK 293T cells. EVs from cells transfected with either Nef^mut^/N, Nef^mut^PL/N or, as control, pVAX1 (Ctrl) were isolated by differential centrifugation, and resuspended in the same volumes of buffer. Preparations were then diluted 20 times and analyzed 5 times with a Nanosight NS300 device. Size distribution and average concentration of EVs are reported as calculated. The results are representative of two independent experiments.

**Figure 4 vaccines-10-01060-f004:**
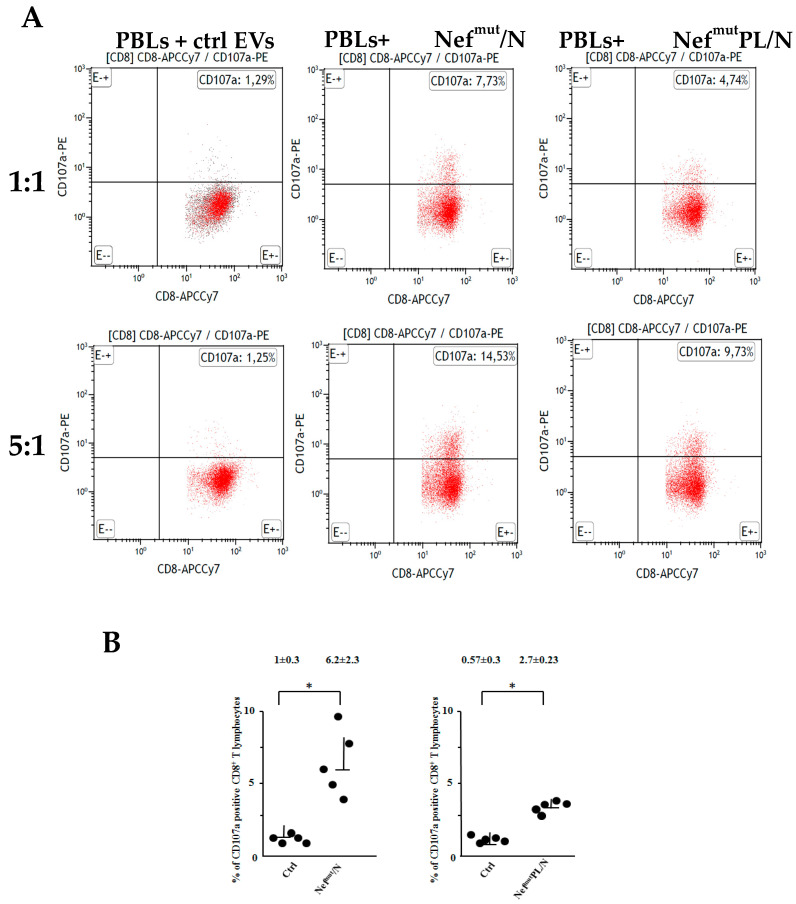
CTL activity as revealed in terms of CD107a expression in CD8^+^ T lymphocytes co-cultivated with DCs challenged with engineered EVs. CD107a FACS analysis on HLA-A.02 PBLs recovered after cross-priming with DCs challenged with EVs isolated from supernatant of cells transfected with pVAX1 (Ctrl), Nef^mut^/N, Nef^mut^PL/N. PBLs were analyzed 5 h after co-culture with HLA-A.02 MCF-7 cells pre-treated with either the unrelated or the SARS-CoV-2-specific peptide. The results in panel (**A**) are representative of five independent experiments carried out with cells from different HLA-A.02 healthy donors. Full data (1:1 cell ratio) are plotted in panel (**B**). Gating strategy (Appendix A): co-cultures labeled with both FITC-conjugated anti-CD3 and APCCy7-conjugated anti-CD8 mAbs were selected for CD3 expression (for PBL identification), and then the CD8^+^ cell sub-populations were identified within CD3^+^ cells. Quadrants were set based on the fluorescence detected after cell labeling with relevant IgG isotypes. * *p* < 0.05.

**Figure 5 vaccines-10-01060-f005:**
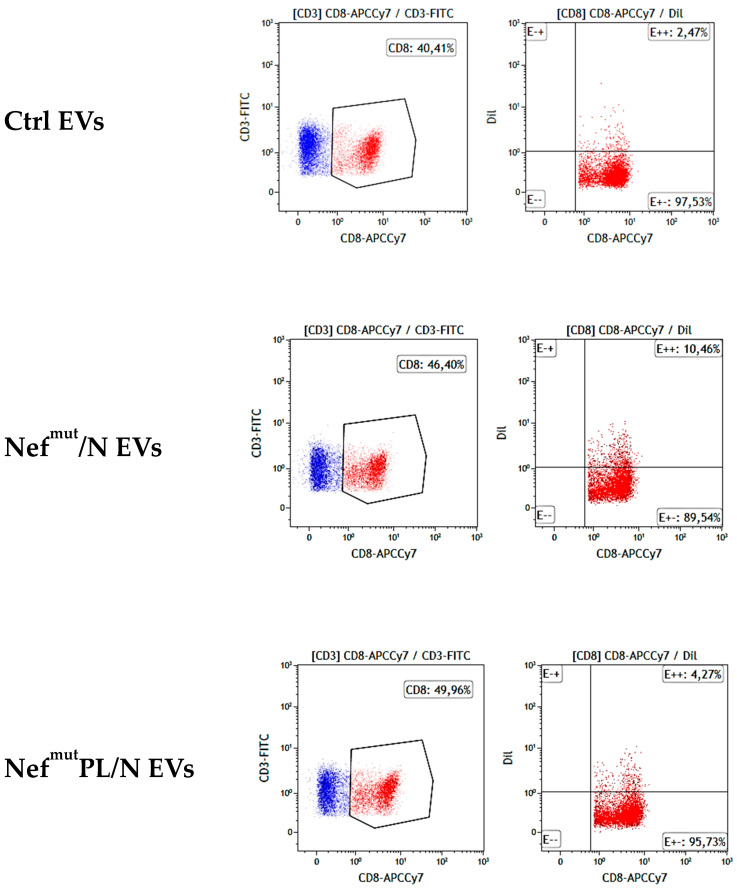
Trogocytosis analysis carried out with CD8^+^ T lymphocytes recovered after cross-priming co-cultures. FACS analysis of PBLs recovered, as described in Figure 4, and co-cultivated with N-peptides treated, CM-Dil-labeled MCF-7 cells. Gating strategy: co-cultures labeled with both FITC-conjugated anti-CD3 and APCCy7-conjugated anti-CD8 mAbs were selected for CD3 expression (for PBL identification), and then the CD8^+^ cells were identified within CD3^+^ cell populations. CD8-labeled cells were finally scored for the CM-Dil-specific fluorescence. Quadrants were set based on the background fluorescence as detected after cell labeling with relevant IgG isotypes. Data are representative of two independent experiments carried out with cells from different HLA-A02 healthy donors.

## Data Availability

The data presented in this study are available on request from the corresponding author. The data are not publicly available due to patent application.

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
