# Peer review of "Activation of Anti-SARS-CoV-2 Human CTLs by Extracellular Vesicles Engineered with the N Viral Protein"

_vaccines, 2022, doi:10.3390/vaccines10071060_

Round 1

Reviewer 1 Report

The manuscript by Manfredi et al. aims at providing the evidence that extracellular vesicles (EVs) engineered with SARS-CoV-2N fused with either Nef^{mut} or its deletion mutant Nef^{mut}PL can elicit anti-SARS-CoV-2 cytotoxic lymphocyte (CTL) immune response in human cells. This is essentially a continuation of previous research. The experimental results are appropriately summarized and illustrated in figures. Al together, they confirm that EVs engineered with Nef^{mut} or Nef^{mut}PL does induce antiviral immunity.

The text is well-written. However, It might be worth providing abbreviation expansion for LPS (lipopolysaccharide) and FACS (fluorescence activated cell sorting), despite they are widely used in the literature.

In my opinion, this paper is suitable for publication in Vaccines.

Author Response

It might be worth providing abbreviation expansion for LPS (lipopolysaccharide) and FACS (fluorescence activated cell sorting), despite they are widely used in the literature.

Abbreviation expansions have been included as required.

Reviewer 2 Report

This article explains an innovative vaccine strategy using extracellular vesicles with fusion viral proteins. There are few points I would like to emphasize that may be useful for the article,

1. Line  31-32 requires references.

2. Line 43-44 requires references.

3.  Line 45-47 requires references.

4.  Figure 1, it would be great if the author includes a western blot with the N protein expression in the cell and EVs.  

5. Figure 1, why the expression of CD81 is different in control EV and other Nef mutant fusion proteins?

6. Figure 1, What is the purpose of those arrows on each side of the blots?

7. Figure 2, the author needs to include a clearer ELISpot picture for the IFN-y responses. Also, the author needs to show whether there is spot formation after the incubation with only Nef mutant protein since immune responses can be mounted against Nef mutant protein as well.

8. Figure 2B, the author needs to mention whether the spot counts are done in per million cells. Also, the IFN-y spots for EVs with Nef mutant protein/N protein are not significantly higher than control.

9. Since this study was performed using fusion protein, there would be CTL activity against both Nef mutant and N protein. In this figure 4, the author need to show the CTL activity for two other proteins i.e. only Nef mutant proteins and only N protein.

10. Is there any changes in MCF-7 proliferation rate during incubation since this could affect the CTL readings.

11. Figure 5, the author could show the trogocytosis for MCF-7 cell upoaded with either Nef mutants or N protein which could explain if there is any improvements in vaccination strategies due to fusion.

Author Response

  1. Line  31-32 requires references.
  2. Line 43-44 requires references.
  3. Line 45-47 requires references.

All required references have been added in the revised ms.

  1. Figure 1, it would be great if the author includes a western blot with the N protein expression in the cell and EVs.  

Our previous investigations led to the conclusion that the fusion with Nefmut is critical for EV incorporation of antigens of interest. This was demonstrated for HPV16-E6 and -E7 (Ferrantelli et al., Cancers, 2021), HER-2 (Anticoli et al., J. Mol. Med., 2018), and GFP (Di Bonito et al. Int. J. Nanomed., 2017). On the other hand, two proteomic-based investigations failed to detect SARS-CoV-2 N-derived products in EVs from infected patients, in the presence, however, of EV-associated Spike-derived products (Barberis et al., Front. Mol. Biosci., 2021; Pesce et al., Front. Immunol. 2022). On this basis, we reasonably assumed that the fusion with Nefmut is mandatory also for the N incorporation into EVs.

  1. Figure 1, why the expression of CD81 is different in control EV and other Nef mutant fusion proteins?

Proteins of the control sample accumulated in the right side of the lane, most likely as consequence of some defects in the relative well. However, such an imperfection did not detract significance to both specificity and quantitative aspects of the assay.

  1. Figure 1, What is the purpose of those arrows on each side of the blots?

As indicated in the figure legend, arrows sign specific products recognized by the antibodies.

  1. Figure 2, the author needs to include a clearer ELISpot picture for the IFN-y responses. Also, the author needs to show whether there is spot formation after the incubation with only Nef mutant protein since immune responses can be mounted against Nef mutant protein as well.

As requested, a clearer EliSpot picture has been included in the revised ms. In addition, data concerning the response to an H2b-specific Nef peptide, which is routinely included in our EliSpot analysis, have been newly included in the revised ms.

  1. Figure 2B, the author needs to mention whether the spot counts are done in per million cells. Also, the IFN-y spots for EVs with Nef mutant protein/N protein are not significantly higher than control.

As reported in the figure legend of the original ms, and newly reported in the y-axis caption of the revised fig. 2B, spot count data refer to 2.5×105 cells. The newly calculated p values (ranging from 0.0179 t0 0.0357) indicate that values obtained with splenocytes from both Nefmut/N and NefmutPL/N-immunized mice were significantly higher than those obtained with splenocytes from control mice.

  1. Since this study was performed using fusion protein, there would be CTL activity against both Nef mutant and N protein. In this figure 4, the author need to show the CTL activity for two other proteins i.e. only Nef mutant proteins and only N protein.

Results depicted in fig. 4B refer to CTL activity specifically directed against target cells binding N peptides on their HLA A.02 Class I MHC. On the other hand, we already demonstrated the induction of a well-detectable CTL activity against Nef-expressing syngeneic cells after cross-priming assays carried out with DCs challenged with Nefmut-incorporating EVs (Anticoli et al., J. Mol. Med., 2018).

  1. Is there any changes in MCF-7 proliferation rate during incubation since this could affect the CTL readings.

Pilot experiments carried out with overnight incubation of MCF-7 cells in the presence or not of N peptides and unprimed PBLs from HLA-A.02 donors did not show significant variation of MCF-7 proliferation (not shown). Hence, we assume that differences in the proliferation rate would not be an issue in the five-hour incubation time needed for the completion of the CD107a assay.

  1. Figure 5, the author could show the trogocytosis for MCF-7 cell upoaded with either Nef mutants or N protein which could explain if there is any improvements in vaccination strategies due to fusion.

As in the case of the assay described in fig. 4, the CTL activity measured in terms of trogocytosis as depicted in fig. 5 refers to the specific recognition by CD8+ T lymphocytes of HLA-A.02-bound N peptides on target cells. This activity is not expected to be influenced by the possible co-existence of Nef-specific CTLs. On the other hand, as argued in point #4, without Nef-fusion N is not expected to be incorporated in EVs to be used in cross-priming experiments.

Reviewer 3 Report

In this study, the author developed a series of engineered exosomes with SARS-CoV-2 N fused at the C-terminus of either Nefmut or a deletion mutant of Nefmut , and then investigated their immunogenicity in mice. They also evaluated the CD8+ T cell function in human dendritic cells after challenging with exosomes including activation-induced degranulation and trogocytosis. However, Study raises many questions that need to be addressed.

Major

1. The major limitation is lack of data regarding the experiments using live viruses in animal models.

2. The author claimed that they assessed the immunogenicity of the exosomes engineered with SARS-CoV-2 N fused with Nef mutant protein in mice through oral administration (line 14). However, they injected the DNA vector vaccine into mouse quadriceps and the mice further underwent electroporation at the site of injection (line 133-134), please explain this.

3. There was no relevance between the sentence “Based on these findings, blocking virus replication at the oral mucosa represents a key step in the fight against respiratory viruses.” (line 43-44) and paragraph 1 (line 31-43). In order to fight against respiratory viruses, intranasal administration induced respiratory mucosa immunity seemed likely to be more effective to control respiratory viruses than oral administration. The advantage of oral administration in the fight against respiratory viruses should be explained.

4. Fig 4 and Fig 5 is not clear, suggest to revise with high resolution images. The red squiggly lines in all the Figure should be delete.

5. Data in Fig 2b need to be statistically evaluated, and the vertical coordinate title should be modified to IFN-γ SFC/10^6 splenocytes.

6. The data in Fig 4b and Figure 5 represented five independent experiments. However, full data have to be given, with single entries as scatter plots. In addition, data need to be statistically evaluated.

7. In this study, the CD8+ T cell function was evaluated by activation-induced degranulation and trogocytosis assay. However, they just measured the CTL reaction in vitro experiment activating human PBMC. The same experiment should be performed in the isolated splenocytes from the vaccinated mice.

8. Furthermore, CD8+ T cell can secret cytokines to indirectly kill cells infected with intracellular pathogens and tumor cells. Cytokines including IFN-γ, IL-2 and TNF-α should be measured in vitro and in vivo experiment.

9. Maturation markers of DC cell including CD40, CD80, CD86 and MHC-1 molecular were suggested to be measured in vitro and in vivo experiment. In addition, cellular uptake of exosomes by DC cells should be performed to determined the cellular uptake efficiency between the full length Nefmut and a deletion mutant of Nefmut.

 10. Though due to limitations of resources regarding conduction of experiments using live viruses and animal models, no protective effect of the vaccine was observed in this study. However, vaccine based on the N protein have been shown to protect mice and macaques. Compare to other studies, the advantage of exosomes vector vaccine should be discussed.

 Minor

11. Please indicate what kind of cell was transfected with the plasmid in method 2.3.

 12. The source of antibodies or reagents were lack of unified. For example, the source of anti-Nef antiserum included company and country (line 119), while the source of PE-conjugated anti-CD107a just contained company (line 183).

13. Flow Cytometry Antibody Information should be given in activation-induced degranulation and trogocytosis assays including cell surface marker.

Author Response

  1. The major limitation is lack of data regarding the experiments using live viruses in animal models.

We agree with this consideration. However, the scope of the present paper was obtaining the proof-of-principle concerning the possibility to translate in humans an innovative model of anti-SARS-CoV-2 immunization. On this basis, “in vivo” experiments using N-engineered EVs in hACE2 mice will be carried out in a short while.

  1. The author claimed that they assessed the immunogenicity of the exosomes engineered with SARS-CoV-2 N fused with Nef mutant protein in mice through oral administration (line 14). However, they injected the DNA vector vaccine into mouse quadriceps and the mice further underwent electroporation at the site of injection (line 133-134), please explain this.

We are sorry for the lack of clarity. The sentence has been modified accordingly. In perspective, the overall scope of our investigations is validating a new method of mucosal anti-SARS-CoV-2 vaccination.

  1. There was no relevance between the sentence “Based on these findings, blocking virus replication at the oral mucosa represents a key step in the fight against respiratory viruses.” (line 43-44) and paragraph 1 (line 31-43). In order to fight against respiratory viruses, intranasal administration induced respiratory mucosa immunity seemed likely to be more effective to control respiratory viruses than oral administration. The advantage of oral administration in the fight against respiratory viruses should be explained.

A more appropriate distinction between oral and intranasal immunizations has been pointed out in the revised sentences.

  • Fig 4 and Fig 5 is not clear, suggest to revise with high resolution images. The red squiggly lines in all the Figure should be delete.

 Images have been ameliorated as requested.

  1. Data in Fig 2b need to be statistically evaluated, and the vertical coordinate title should be modified to IFN-γ SFC/10^6 splenocytes.

As requested, data in fig. 2b have been statistically evaluated, and the y-axis labeled appropriately.

  1. The data in Fig 4b and Figure 5 represented five independent experiments. However, full data have to be given, with single entries as scatter plots. In addition, data need to be statistically evaluated.

The referee’s request applies for data reported on fig. 4, which has been modified accordingly. Conversely, as stated in the figure legend of the original ms, data reported in fig. 5 are representative of two independent assays.

  1. In this study, the CD8+ T cell function was evaluated by activation-induced degranulation and trogocytosis assay. However, they just measured the CTL reaction in vitro experiment activating human PBMC. The same experiment should be performed in the isolated splenocytes from the vaccinated mice.

      The ability of the Nefmut-based vaccine platform to induce antigen-specific CTL activity within splenocytes of vaccinated mice has been reproducibly observed for a plethora of viral antigens (Di Bonito et al., Int. J. Nanomed., 2017; Anticoli et al., Biotechnol. J., 2018). Conversely, focus of the present ms is demonstrating the possibility to translate into the human system what we previously observed in vaccinated mice in terms of SARS-CoV-2 N-specific CTL activity (Ferrantelli et al., Vaccines 2021; Ferrantelli et al:, Viruses, 2022).

  1.   Furthermore, CD8+ T cell can secret cytokines to indirectly kill cells infected with intracellular pathogens and tumor cells. Cytokines including IFN-γ, IL-2 and TNF-α should be measured in vitro and in vivo experiment.

The detection of N-specific polyfunctional CD8+ T lymphocytes in splenocytes as well as in cells isolated from BALFs of Nefmut/N vaccinated mice has been recently described (Ferrantelli et al. Viruses, 2022). These results confirmed previously published data obtained in mice vaccinated with Nefmut/HPV16-E6 and/or –E7 expressing vectors (Ferrantelli et al, Cancers, 2021).

  1. Maturation markers of DC cell including CD40, CD80, CD86 and MHC-1 molecular were suggested to be measured in vitro and in vivo experiment. In addition, cellular uptake of exosomes by DC cells should be performed to determined the cellular uptake efficiency between the full length Nefmutand a deletion mutant of Nefmut.

      Both differentiation and maturation markers are routinely tested to assess the proper in vitro generation of immature/mature DCs. A representative analysis is reported on Supplementary Figure 3 of the revised ms.

      We already reported data on DC internalization of engineered EVs (Anticoli et al, J. Mol. Med., 2018). These analyses were carried out by exploiting the strong fluorescence signal generated by the Nefmut-GFP fusion product incorporated into EVs. In light of the not adequate sensitivity/specificity of commercially available anti-Nef and anti-N antibodies, such an analysis appears at the present not feasible using EVs incorporating either Nefmut/N or NefmutPL/N fusion proteins.

  1. Though due to limitations of resources regarding conduction of experiments using live viruses and animal models, no protective effect of the vaccine was observed in this study. However, vaccine based on the N protein have been shown to protect mice and macaques. Compare to other studies, the advantage of exosomes vector vaccine should be discussed.

In this ms we propose N-engineered EVs as a prophylactic treatment at the port of entry of SARS-CoV-2 infection. This strategy differs from that based on EVs generated by injection of Nefmut-based DNA vectors, whose advantages have been discussed elsewhere (Ferrantelli et al., Viruses, 2022). Possible advantages of the strategy described in the present paper include recognition of all variants, expectedly very low adverse events, immunity at the viral port of entry, and, on the basis of data published on SARS-CoV survivors (Le Bert et al., Nature 2020), long lasting immunity. These concepts have been highlighted in the “Discussion” section of the revised ms.

 Minor

  1. 11.  Please indicate what kind of cell was transfected with the plasmid in method 2.3.

The transfected cell type has been indicated in the revised ms.

  1. The source of antibodies or reagents were lack of unified. For example, the source of anti-Nef antiserum included company and country (line 119), while the source of PE-conjugated anti-CD107a just contained company (line 183).

These requests have been fulfilled in the revised ms.

  1. Flow Cytometry Antibody Information should be given in activation-induced degranulation and trogocytosis assays including cell surface marker.

As requested, info on all used antibodies have been included in the revised ms.

Round 2

Reviewer 2 Report

I feel all the points have been addressed in the paper and can be accepted for publication.